# Spectrum of Disease Manifestations in Patients with Selective Immunoglobulin E Deficiency

**DOI:** 10.3390/jcm10184160

**Published:** 2021-09-15

**Authors:** César Picado, Iñaki Ortiz de Landazuri, Alexandru Vlagea, Irina Bobolea, Ebymar Arismendi, Rosanel Amaro, Jacobo Sellarés, Joan Bartra, Raimon Sanmarti, José Hernandez-Rodriguez, José-Manuel Mascaró, Jordi Colmenero, Eva C. Vaquero, Mariona Pascal

**Affiliations:** 1Institut Clinic Respiratory, Hospital Clinic, Universitat de Barcelona, 08036 Barcelona, Spain; Bobolea@clinic.cat (I.B.); earismen@clinic.cat (E.A.); ramaro@clinic.cat (R.A.); sellares@clinic.cat (J.S.); JBARTRA@clinic.cat (J.B.); 2Institut d’Investigacions Biomèdiques August Pi i Sunyer (IDIBAPS), 08036 Barcelona, Spain; sanmarti@clinic.cat (R.S.); jhernan@clinic.cat (J.H.-R.); mascaro@clinic.cat (J.-M.M.); jcolme@clinic.cat (J.C.); evaquero@clinic.cat (E.C.V.); MPASCAL@clinic.cat (M.P.); 3Centro de Investigaciones Biomédicas en Red de Enfermedades Respiratorias (CIBERES), 28029 Madrid, Spain; 4Immunology Department, CDB. Hospital Clinic, Universitat de Barcelona, 08036 Barcelona, Spain; ortizdelan@clinic.cat (I.O.d.L.); vlagea@clinic.cat (A.V.); 5Department of Rheumatology, Hospital Clinic, Universitat de Barcelona, 08036 Barcelona, Spain; 6Department of Autoimmune Diseases, Hospital Clinic, Universitat de Barcelona, 08036 Barcelona, Spain; 7Department of Dermatology, Hospital Clinic, Universitat de Barcelona, 08036 Barcelona, Spain; 8Liver Unit, Hospital Clinic, Universitat de Barcelona, 08036 Barcelona, Spain; 9Centro de Investigaciones en Red de Enfermedades Hepáticas y Digestivas (CIBEREHD), 28029 Madrid, Spain; 10Department of Gastroenterology, Hospital Clinic, Universitat de Barcelona, 08036 Barcelona, Spain

**Keywords:** autoimmune diseases, Immunoglobulin E, Immunoglobulin deficiency, infections, malignancy

## Abstract

Background: Selective IgE deficiency (SIgED) has been previously evaluated in selected patients from allergy units. This study investigates the effects of SIgED on the entire population in a hospital setting and sought to delineate in detail the clinical aspects of SIgED. Methods: A retrospective study of the data obtained from electronic medical records of 52 adult patients (56% female) with a mean age of 43 years and IgE levels of <2.0 kU/L with normal immunoglobulin (Ig) IgG, IgA, and IgM levels, seen at our hospital, without selection bias, from 2010 to 2019. Results: Recurrent upper respiratory infections were recorded in 18 (34.6%) patients, pneumonia was recorded in 16 (30.7%) patients, bronchiectasis was recorded in 16 (30.7%) patients, and asthma was recorded in 10 (19.2%) patients. Eighteen patients (34.6%) suffered autoimmune clinical manifestations either isolated (19%) or combining two or more diseases (15%), Hashimoto’s thyroiditis being the most frequent (19%), which was followed by arthritis (10%) and thrombocytopenia and/or neutropenia (5.7%). Other less frequent associations were Graves’ disease, primary sclerosing cholangitis, Sjögren’s syndrome, and autoimmune hepatitis. Eczematous dermatitis (15.3%), chronic spontaneous urticaria (17.3%), and symptoms of enteropathy (21%) were also highly prevalent. Thirty percent of patients developed malignancies, with non-Hodgkin lymphomas (13.4%) being the most prevalent. Conclusions: The clinical manifestations of SIgED encompass a variety of infectious, non-infectious complications, and malignancy. Since it cannot be ruled out that some type of selection bias occurred in the routine assessment of IgE serum Ievels, prospective studies are required to better characterize SIgED and to determine whether it should be added to the list of antibody deficiencies.

## 1. Introduction

A recently updated classification distributes innate errors of immunity into 10 groups, one of which is considered to be due to “antibody deficiencies” (Group 3) [1,2].

Immunoglobulins A (IgA), M (IgM), and G (IgG) are central in the humoral immune response and play a fundamental role in protecting against infections caused by all kinds of agents (viruses, bacteria, protozoa, parasites), and they represent the defense mediated by antibodies, which are part of the so-called acquired immunity [3].

Immunoglobulin E (IgE) has been conventionally related to the immune response against helminth infection, and its levels are particularly high in patients who suffer from a parasitic infestation [4]. IgE is also involved in type I hypersensitivity allergic reactions, which are diseases where it is also common to find high levels of specific IgE against allergens. Most IgE is found bound to its high-affinity receptor FcϵRI located on the surface of mast cells and basophils. The binding of the allergen to the specific IgE/FcϵRI complex triggers the degranulation of mast cells and basophils that release numerous substances (vasoactive, bronchoconstrictors, interleukins), which are ultimately responsible for the clinical manifestations of the allergic response (rhinitis, asthma, urticaria, angioedema, anaphylaxis) [5].

Various types of immunodeficiencies associated with low levels of one or a combination of IgA, IgG, and IgM immunoglobulins are recognized. [1,2]. The most studied combined form is known as “common variable immunodeficiency” (CVID), which is a disorder characterized by reduced serum levels of IgG, which can be combined with a reduction of IgA or IgM, or both, which is associated with recurrent sinopulmonary infections, autoimmune disorders, granulomatous diseases, and increased risk of malignancy and altered response of antibodies against infections [6,7].

Selective IgG deficiency (SIgGD) encompasses any subject with a serum IgG level below normal range with normal IgA and IgM levels. Studies comparing the SigGD and CVID patients found that the CVID group was more likely to have bronchiectasis, poorer responses to vaccines, and a higher incidence of autoimmune cytopenias, granulomas, splenomegaly, and lymphoid neoplasms than those with SigGD [8].

IgG subclass deficiency (IgGSD) is a heterogeneous subtype of primary immunodeficiency, which is defined as the triad of frequent or severe respiratory tract infections, subnormal levels of one or more of the four IgG subclasses, and decreased IgG response to pneumococcal polysaccharides. Many adults with IgGSD also have autoimmune conditions or atopy [9,10].

Selective IgA deficiency (SIgAD) [11,12], and selective IgM (SIgMD) [13,14] are diagnosed in a diverse group of patients, ranging from completely asymptomatic individuals to people with recurrent infections, allergic diseases, autoimmune processes, and malignant tumors.

The question is: are there any similar diseases associated with selective IgE deficiency (SIgED)? Conventionally, normal serum IgE values are considered to range between the technical detection limit (≤2 kU/L) and up to 100 kU/L. An excess of IgE (>100 kU/L) can be established but, in contrast to the other immunoglobulins, there is no generally accepted minimum level to establish an IgE deficiency. In various studies in the literature, different cut-off points have been used to define IgE deficiency [15,16,17,18,19]. Most clinicians do not attribute any pathological significance to very low IgE values, even those that are unquantifiable (≤2 kU/L), which are usually considered as “normal”.

Low IgE is frequently associated with deficiencies in other immunoglobulins, particularly in patients with CVID [20,21,22]. Based on this observation, the use of routine IgE measurement has been proposed as the first step to detect the presence of CVID [21,22].

In the classification of primary immunodeficiencies attributed to antibody deficiency, the presence of low IgE values is mentioned, but it is always associated with deficiencies in some of the other immunoglobulins [1,2]. The possibility of an immunodeficiency associated only with an SIgED is not considered in the classification. However, a few studies have reported that an SIgED may be the biomarker of an immunodeficiency with a significant clinical impact that has been overlooked until now [23,24,25,26].

The studies analyzing the potential role of SIgED are retrospective and include a limited number of cases. Furthermore, most of the patients included in these studies were selected from allergy units [23,26] or were patients having any allergy-related symptoms and/or requesting antiallergy medications [25], which is a bias that could have limited the spectrum of diseases found associated with SIgED. Despite these limitations, it is worth noting that these studies show that individuals with a low level of IgE, with normal values for the other immunoglobulins, present recurrent respiratory infections, suffer from autoimmune diseases, and upper and lower airway diseases [23,24,25,26], similar to those described in patients with CVID [6,7,8], IgGSD [9] or with SIgGD [8], SIgAD [11,12], and SIgM D [13,14].

The predisposition to develop neoplasms in patients with antibody deficiency, either in combination or due to selective deficits of IgA or IgM, is widely documented [27]. In the same way, the scientific information supports that IgE deficiency is a predisposing factor for the development of malignancies [28].

The hypothesis of this study establishes that isolated IgE deficiency is associated with diseases similar to that described in other antibody deficiencies, but its clinical spectrum has been underestimated.

This study is the first to research the effects of SIgED in the entire population in a hospital setting with a 2-year follow up and sought to delineate in detail the clinical aspects of SIgED.

## 2. Patients and Methods

Any patient who was found to have an IgE concentration ≤2 kU/L with normal IgG, IgM, and IgA concentrations with at least 2-year follow up at our institution between January 2010 and December 2019 was included in the study. A total of 151 patients were analyzed, of whom 99 were excluded for different reasons shown in Figure 1. The remaining 52 patients with SIgED and regular follow-up in the hospital were included in the study. Medical records were reviewed and discussed together with the various specialists involved in their routine care. Of the 52 patients, 31 were female (56%), and the mean age was 43 years (range 18–87).

In patients with a suspicion of either a respiratory or food allergy, we routinely perform skin prick tests (SPTs) in our institution with a panel of commercial allergenic extracts of the most prevalent aeroallergens and food allergens in our area (Laboratorios LETI, Madrid, Spain). Foods suspected by the clinical history and not included in the standard panel are also usually tested with a commercial extract if available, or by prick-prick according to standard methods. Serum levels of IgG, IgA, IgM, and IgGs were measured by immunoturbidimetry (Atellica NEPH 630 Solution System. Siemens Healthineers, Germany). Serum total and specific IgE levels were measured by immunofluorescence enzyme immunoassay (ImmunoCAP, ThermoFisherScientific, Uppsala, Sweden). Serum IgA, IgM, and IgG values of the participants were: IgM 1.10 (range, 0.41–2.42) g/L (normal values (0.36–2.6 g/L); IgG 10.7 g/L (range, 7.10–13.30) g/L; and IgA 2.12 g/L (range 0.95–4.80) g/L (normal values 0.66–3.6 g/L). The study was approved by the Ethics Committee of the Hospital Clinic (Ethical Code: HCB/2021/0758).

## 3. Results

### 3.1. Infections

Three or more yearly upper respiratory infections (URI) (rhinorrhea, nasal congestion, and productive cough), requiring antibiotic therapy for at least two consecutive years, were recorded in 18 (34.6%) of the SIgED patients. Sixteen patients (30.7%) had suffered one or more episodes of pneumonia (range 1 to 3). Median IgG, IgM, and IgA levels were not significantly different in patients with or without URI or pneumonia (data not shown). In two patients, chronic respiratory infection with *M. avium* complex was identified. Recurrent episodes of otitis were recorded in four patients. Three patients had suffered from herpesvirus infections, and one had suffered from chronic pyelonephritis.

### 3.2. Lung Diseases

Twenty-nine patients underwent chest computed tomography (CT) scanning. The radiological study demonstrated the presence of bronchiectasis in 16 patients (30.7% of total sample) affecting between one and three lobes and mostly cylindrical, peribronchial thickening in three, air trapping in two, atelectasis in three, micronodules in three, cyst in three, pulmonary emphysema in three (all ex-smokers), ground glass opacities in six, and interstitial lung fibrosis in one. The chest CT scan was considered normal in six patients (all of them suffering from frequent respiratory infections). In two patients, the combination of chest CT findings (micronodules, cysts), bronchoalveolar lavage (BAL) fluid results (lymphocytic inflammation and multinucleated giant cell), and the lung biopsy of a nodule (lymphocytic infiltration) indicated the presence of lymphocytic interstitial lung disease. Ten (19.2%) patients were diagnosed with asthma with different levels of severity. Eight patients (15.3%), six of them associated with asthma, referred symptoms of allergic rhinitis, and four of them reported clinical symptoms apparently exacerbated seasonally (spring, autumn). In all patients, SPTs and specific IgE for common allergens) were negative (Table 1).

### 3.3. Autoimmune Diseases

Eighteen patients (34.6%) suffered autoimmune clinical manifestations, either isolated (19%) or combining two o more diseases (15%). Hypothyroidism was diagnosed in 10 patients (19.2%), eight secondary to Hashimoto’s thyroiditis, and two resulted from previously treated hyperthyroidism (Graves’ disease). Other less frequently found autoimmune diseases are shown in Table 1.

### 3.4. Gastrointestinal and Liver Diseases

Symptoms of enteropathy such as intermittent or persistent chronic diarrhea, abdominal pain, and bloating were present in 11 patients (21%). Some patients associated their symptoms with the ingestion of certain foods. In all cases, both allergen SPTs and specific IgE studies with the putative culprit foods were negative. Fructose and lactose intolerance were assessed in four patients, and only one tested positive in the lactose test. Celiac disease was excluded in most (nine patients) but not all patients by anti-transglutaminase IgG serology. A gluten-free diet was tested in four patients with inconsistent or negative symptomatic response. Biopsies of colon mucosa and/or small intestine were obtained in six patients, and the histological findings were: intraepithelial lymphocytosis (four patients), lymphoid hyperplasia forming aggregates (one patient), and enteritis with chronic inflammation, eosinophilic infiltration, crypt distortion, and gland destruction (one patient). Acute severe autoimmune hepatitis was diagnosed in one patient who had required two liver transplantations. Two patients suffered from primary sclerosing cholangitis (PSC) progressing to cirrhosis requiring liver transplantation. The two patients also suffered from ulcerative colitis (Table 1).

### 3.5. Cutaneous Findings

Eczematous dermatitis (eight patients, 15.3%) associated with moderate or severe itching in most cases, chronic spontaneous urticaria (CSU) (nine patients, 17.3%), angiedema (four patients associated with CSU), and chronic leg ulcers (two patients) were present among the SIgED patients (Table 1).

### 3.6. Tumours

Sixteen (30.7%) patients developed malignancies including non-Hodgkin lymphomas (seven patients, 13.4%), chronic lymphocytic leukemia (two patients, one evolving from a lymphoma), and various types of malignant and non-malignant tumors, as shown in Table 2. Four patients developed more than one tumor.

### 3.7. Other

Mastocytosis (two patients), monoclonal gammopathy of undetermined significance (MGUS) (two patients), fatigue (seven patients, 13.4%), and polyarthralgia (nine patients, 17.3%) were also reported by some patients as major complaints. Follicular hyperplasia affecting lymph nodes located in the mediastinum, armpits, groin, supraclavicular area, or abdomen were found in eight (15.3%) patients, which, when biopsied (three patients), showed a pattern of non-specific lymphoid reactivity, although one of them later evolved to a lymphoma. Arterial hypertension (13 patients, 33.3%) and ischemic heart disease (four patients, 7.7) were also documented.

### 3.8. SIgED and IgG Subclasses

Serum levels of IgG1, IgG2, IgG3, and IgG4 had been assessed in 14 patients and were normal in all but three patients: one with low IgG3 and two with low IgG4 (data not shown).

## 4. Discussion

It is generally accepted that low levels of IgA, IgM, and IgG predispose to respiratory bacterial and viral infections. The high incidence of URI and pneumonia in our patients with SIgED is in keeping with that reported for CVID [8,9,29], SIgGD [8], IgGSD [9,10], SIgAD [11,12], and SIgMD [13,14]. The mechanism by which an SIgED may also predispose to lung infections remains to be elucidated. IgE is usually related to protection against parasites [4], but its role in other infections is not usually considered, despite there being studies that have demonstrated the presence of specific IgE antibodies against viruses such as H1N1 influenza [30], respiratory syncytial [31], HIV1 [32], varicella [33], parvovirusB19 [34], and rhinovirus (RV) [35]. Anti-HIV1 IgE has been shown to inhibit HIV1 production in infected cell culture, the inhibitory effect being reversed when IgE was removed from the culture [32]. In a study involving children with HIV-1 infection, opportunistic infections were less frequent in children with high serum IgE levels than in those with low IgE levels [36].

It is generally assumed that IgE does not play any relevant role in the immune response against bacteria. However, there are studies reporting that IgE antibodies provide immunity against bacteria such as *Borrelia burgdorferi* [37]. It was recently discovered that the antibacterial activity of mast cells against *Staphylococcus aureus* (SA) in mice was markedly enhanced by the presence of IgE directed against bacterial components. Animal models deficient in IgE or FcϵRI were unable to mount protective immune responses against SA infections [38]. Furthermore, other authors have found that SIgED deficiency predisposes to recurrent upper and lower airways with common respiratory bacteria such as *Haemophilus influenza, Moraxella catarrhalis*, and *Streptococcus pneumoniae* [23].

Taken together, these findings support the notion that a reduced synthesis of IgE may result in an immunodeficient response against virus and bacteria. As far as we know, the response to vaccines of patients with SIgED has never been studied.

Two of our patients (3.8%) had chronic *Mycobacterium avium* infection. The patients had clinical and radiological findings suggestive of ‘Lady Windermere Syndrome’ (LWS), which is characterized by chronic bronchiectasis in slender women, with scoliosis and/or *pectus excavatum*, and chronic productive cough. Multigenic variants with potential defects in proteins encoded by various genes might contribute to LWS by reducing both IFN-γ production and increasing transforming growth factor (TGF)-β levels in response to non-tuberculous mycobacterium (NTM) [39,40,41,42]. NTM infection has been reported in a very small percentage of patients with CVID (0–1%) [43]. Chronic respiratory infection with *M. avium* complex and bronchiectasis were identified in 5% of patients with SIgAD [44]. So far, in patients with SIgAD, SIgGD, and IgGSD, no NTM infections have been reported. The potential role of SIgED in NTM infection is unknown and should be evaluated in a larger series of patients with lungs infected with these pathogens.

It is generally assumed that in CVID patients, recurrent airway infections and persistent airway inflammation can lead to a vicious circle airway remodeling process resulting in bronchiectasis [43]. A recent analysis of existing data on the clinical presentation of CVID found that bronchiectasis was present in the CT scan in almost one-third of patients (28%, 95% CI 18–40) [29]. Bronchiectasis has been found in up to 14% of SIgAD patients and is more commonly reported when associated with IgG subclass deficiency [44,45,46]. We found that bronchiectasis was present in 30.7% of our SIgED patients, which is a percentage similar to that reported in CVID, which is an observation that suggests that the lack of IgE has a significant negative impact on the immune defense mechanisms of the lung. In contrast to our findings, bronchiectasis is not even mentioned in the few studies reporting the clinical manifestations present in SIgED patients, which is most probably due to the lack of CT scan evaluation in patients with frequent respiratory infections [23,24,25,26].

Viral infections are a strong risk factor for developing asthma in children, and they are major contributors to exacerbations of asthma in both children and adults [47]. The link between viruses and upper (rhinitis) and lower respiratory diseases (asthma) might explain the high percentage of patients with CVID that are diagnosed with asthma (25%, 95% CI 17–35) and rhinitis (18%, 95% CI 8–31) [29]. What is not yet clear is the mechanism underlying this association. Are CVID patients with asthma-like clinical symptoms a distinct hyperreactive airway phenotype? Or, are they subjects to genetic factors predisposing them to develop asthma, which is unmasked early by the presence of the immunodeficiency? Mutations in the TNFRSF13B gene have been found in CVID patients [48] and are also associated with an increased risk of asthma development [49].

CVID patients with asthma and rhinitis are often clinically characterized as allergic [29,43]. Interestingly, some of our patients reported nasal and bronchial allergic-like reactions—a few of the associated with seasonal exacerbations. This is not surprising, given the presence of ultralow serum levels of IgE, SPTs, and that the in vitro tests for serum-specific IgE against common allergens were negative in all patients. It is theoretically plausible that allergic-type symptoms could be due to the presence of IgE in the respiratory tract, which is something similar to so-called local allergic rhinitis (LAR) and local allergic asthma (LAA) [50,51]. These diseases are characterized by the negativity of the skin-prick test and serum-specific IgE for all relevant aeroallergens in a patient with upper and lower airway symptoms suggestive of allergy, and who tested positive in the nasal and bronchial allergen challenge [50,51]. Although the cells and the main sites of IgE production in humans remains to be fully characterized, it is assumed that IgE is produced in the peripheral blood and locally in various tissues, including the nose and lung [52]. One may speculate that airway mucosal IgE in patients with SIgED is still capable of developing respiratory allergic responses in a similar way to that described in LAR and LAA [51,52]. Interestingly, patients with CVID, IgE deficiency, and a history suggestive of allergic asthma with negative allergen SPTs did not show any bronchial reactions when subjected to an allergen challenge, but the exposure to allergens increased the airway response to histamine [53]. Whether the acquired airway hyperresponsiveness was due to a local IgE-dependent or another non-IgE related mechanism remains to be clarified. It is also unclear whether allergen-induced hyperresponsiveness can indirectly account for the symptoms of those IgE-deficient patients associated with allergen exposure. Allergy symptoms may have been confused with unspecific airway hyperreactivity-related clinical manifestations.

Similar to CVID, SIgAD has also been associated with allergic rhinoconjunctivitis and asthma [44,54,55]. However, the prevalence of these diseases shows large differences among studies, ranging from 13% [44] to 83% [55]. Furthermore, one age- and gender–matched survey found an increased prevalence of allergic rhinoconjunctivitis, but no differences were found in asthma prevalence between SIgAD and controls [55]. Thirty five percent of patients with SIgMD had atopic diseases, including allergic rhinitis and asthma [45], while allergic asthma and/or allergic rhinitis were the second commonest manifestations in patients with IgGSD without any subclass predominance [56].

In previous studies, SIgED has been found to be associated with a higher prevalence of non-allergic reactive airways disease (rhinorrhea, nasal congestion, dry cough, and/or wheezing) (73%) compared with controls (20%) [23], and with asthma or hyperreactive airway disease (26.5% vs. controls 6.8%) in children but not in adults [25]. The prevalence of asthma (19.2%) found in our study was higher than that reported in the adult Spanish population (range 10–16.7%) [57].

Interstitial lung disease (ILD) is a frequent (15–60%) non-infectious complication of CVID [58]. The histology of ILD in CVID shows heterogeneous and often mixed patterns, including lymphoid hyperplasia, lymphoid interstitial pneumonitis, follicular bronchiolitis, non-necrotizing granulomatous inflammation, organizing pneumonia, and interstitial fibrosis [59]. Granulomatous-lymphocytic interstitial lung disease (GLILD) is often used as a term to describe ILD with lymphocytic infiltrates and/or granulomata in CVID [59]. However, not all ILD in CVID have pulmonary granulomata, and therefore, the term does not fully cover the heterogeneous spectrum of the histopathology found in lung samples from CVID patients [59]. Approximately 20% of patients with ILD present polyclonal lymphocytic infiltration or non-malignant hyperplasia of the lymph nodes in addition to granuloma [60]. Monogenic disorders causing CVID-like diseases have also been reported in patients with ILD [61,62,63]. Patients with ILD have distinct clinical and immunological phenotypes in keeping with immune dysregulation, in contrast to those without ILD or those with bronchiectasis alone [64]. Recent studies have shown that ILD is also present in the lung of patients with selective immunoglobulin deficiencies, including SIgAD, SIgGD, and IgGSD, with a pattern of lymphoid proliferation and granulomata identical to that found in CVID [65,66,67]. Lung biopsies from CVID patients usually show some degree of fibrosis, which can be extensive, and is the predominant finding in up to 6.5% of cases [64]. The presence of extensive lung fibrosis is associated with a poor prognosis [68].

In our study, we found two patients with radiological, BAL fluid cytology, and histological lung findings suggestive of ILD, and one patient with clinical and radiological findings commonly associated with severe interstitial lung fibrosis, which caused her death. Although not confirmed by biopsy, in four patients, the CT scan showed ground-glass opacities, pulmonary nodules, and mediastinal lymphadenopathy, which are images considered highly suggestive of ILD [64]. Taken together, our observations suggest adding SIgED to the immunodeficiencies potentially associated with ILD.

A substantial number of CVID patients (27%, 95% CI 22–32%) develop autoimmune manifestations [29]. Studies have shown that SIgGD [9,12] SIgMD [13,45,69], IgGSD [56], and SIgAD [11,12,44,55] are also associated with systemic and organ-specific autoimmune diseases. The clinical spectrum of autoimmunity in CVID and other selective immunodeficiencies is very wide and includes a plethora of hematologic (cytopenia, thrombocytopenic purpura, hemolytic anemia, Evans syndrome), and non-hematologic diseases (autoimmune thyroid diseases, rheumatoid arthritis, unspecific inflammatory arthritis, Sjögren´s syndrome, systemic lupus erythematous (SLE), autoimmune hepatitis) [65]. In our study, we found that SIgED was associated with hematologic and non-hematologic autoimmune diseases, with percentages similar to those described in other immunodeficiencies. Isolated and mixed autoimmune diseases were also significantly more frequent in adults and children with SIgED compared with control populations in previous studies [23,25]. As in our study, thyroid diseases (Hashimoto’s thyroiditis and Grave’s disease), cytopenias, SLE, and arthritis were autoimmune diseases reported in patients with SIgED [23,25]. Taken together, these findings support that autoimmunity is a relevant component of the clinical presentation of SIgED.

CSU, in some cases associated with angioedema, was frequently diagnosed in our patients. In contrast, CSU and angioedema are not usually listed among the more common clinical manifestations in CVID [29,65]. However, some reports point out that we should not overlook the association of CSU with CVID [70,71,72,73]. CSU has been found in 4.9% of patients with SIgAD compared with 0.9% in controls [74], and in up to 12% of patients with SIgMD [75]. A statistically significant prevalence of CSU was observed in patients with SIgED (19%) compared with controls (0.8%) in one study [25], while another study did not find any differences between patients (11%) and controls (11%) [23].

Eczematous dermatitis was also found in a high percentage of our patients. The “eczema group” is frequently (33.7%) diagnosed in patients with various primary immunodeficiencies [76], but it is not included among the most common manifestations of patients with CVID [29,64]. Interestingly, severe eczematous dermatitis is characteristic of diseases of the immune system associated with both autosomal dominant and autosomal recessive forms of hyper IgE syndrome [77]. In some of our patients, eczematous dermatitis was associated with severe itching requiring regular treatment with oral corticosteroids, in some cases complemented with immunosuppressive therapy. A previous study in patients with SIgED could not find any differences in skin rash complaints between patients and controls [23], while in another study, rashes diagnosed as psoriasis and seborrheic dermatitis were found to be significantly higher in SIgED patients than in controls [25].

Gastrointestinal symptoms that may mimic inflammatory bowel disease are very frequent in patients with CVID. Intermittent or persistent diarrhea (27%, 95% CI 21–34) [29,78], bloating (34%) [78], and abdominal pain (26%) [78], are the most common gastrointestinal symptoms. The enteropathy of CVID may affect any part of the gastrointestinal tract and is associated with various histological findings, including intraepithelial lymphocytosis (46%), a decreased number of plasma cells in the GI tract mucosa (62%), and lymphoid hyperplasia (38%) [78]. Many other histological findings, such as eosinophilic or lymphocytic enteritis, villous atrophy, collagenous enteritis, and granulomatous inflammation are less frequently found in biopsies [29,78]. Studies in patients with SIgMD show great variability in the prevalence of gastrointestinal manifestations, without clearly differentiating those that may be due to an enteropathy similar to that found in CVID [14,44,55,75]. Both chronic and recurrent diarrhea are more common among individuals with SigAD than in the control population. However, many of these cases are associated with either celiac disease or inflammatory bowel disease [56]. A comparison study shows more biopsy-confirmed enteropathy cases among CVID patients (7%) than among SigGD patients (3.2%) [8]. Gastrointestinal symptoms have not been reported associated with IgGSD [56]. Eleven (21%) of our patients reported gastrointestinal symptoms suggestive of enteropathy. However, only five had been assessed by endoscopy (9.6%), but all had histological findings (intraepithelial lymphocytosis, lymphoid hyperplasia, lymphocytic and eosinophilic enteritis) usually found in the enteropathy of CVID. Previous studies in SigED patients offer scant data on gastrointestinal symptoms, although they mention that some patients had been diagnosed with food allergy in some children [25], and inflammatory bowel disease and celiac disease in some adults [23], but without data from histological studies.

Liver diseases have been reported in up to 12.7% of CVID patients [67], ranging from elevated alkaline phosphatase to nodular regenerative hyperplasia (NRH), autoimmune hepatitis, liver cirrhosis, and primary sclerosing cholangitis (PSC) [79]. Some isolated cases of liver disease with NRH, and acute autoimmune hepatitis have been reported in patients with SIgAD [80] and SIgMD [81,82]. Acute severe autoimmune hepatitis (one patient) and PSC (two patients) progressing to cirrhosis were diagnosed in our SIgED patients. The three patients required liver transplantation. The two patients with PSC suffered from ulcerative colitis [83], and one of them developed a cholangiocarcinoma, which are both entities considered common complications in PSC [84].

Interestingly, high serum levels of IgE have been found associated with a lower incidence of biliary carcinoma in patients with PSC [85]. This is not an unexpected finding, since numerous epidemiological studies carried out in recent years have shown an inverse relationship between elevated IgE levels and malignant processes [28,86,87,88,89]. In keeping with these epidemiological observations, we found a strikingly high prevalence of malignancies (30%), with non-Hodgkin lymphomas (13.4%) topping the list in our patients. These findings are also very similar to those reported in some previous publications of SIgED patients, where a significant increase in malignant processes (lymphomas, lymphocytic leukemia, and epithelial cancers) was observed compared with the control group [25]. The high frequency of malignant processes in our study supports the relevant role played by IgE in antitumor surveillance detected in epidemiological studies.

The association between immunodeficiencies and cancer is well established. In CVID patients, the most commonly reported malignancies are non-Hodgkin lymphomas and various solid cancers (breast, colon, lung, gastric, ovarian, melanoma) [64,90,91]. A recent study has shown that the link between IgE and malignancies appears to be specific and independent of the presence of CVID in patients with IgE deficiency, which is a finding that lends further support to IgE’s leading role in cancer development [92]. IgA deficiency is also associated with a moderately increased risk of cancer, with excess risks of gastrointestinal cancer not related to the presence of celiac disease [93]. Various types of cancer have been found in patients with SIgMD in some [44,69,94] but not all studies [94,95].

Fatigue as a major complaint was present in a high percentage (13.4%) of our patients. Chronic fatigue was also found significantly more frequently in patients with SIgED than in controls in previous studies, 3.8% vs. 0.3% [25] and 30% vs. 4% [23]. Fatigue is a very common complaint in patients with primary immunodeficiency disorders [96], particularly in patients with CVID (40%) [29], but it has only been reported in isolated patients with other selective immunoglobulin deficiencies. The prevalence of polyarthralgias in our patients was high (17.3%), but it was even less frequent than previously reported in patients with SIgED (32% vs. 7% in controls) [23]. SIgED associated with both arterial hypertension (37.7%) and ischemic heart disease (25.2%) was previously reported by E Magen et al. [97]. We found a similar prevalence of arterial hypertension (33.3%) but lower ischemic heart disease (7.7%). The mechanisms involved in these associations remain to be elucidated [97]. The same group has also reported that in comparison to a control group, a significantly larger proportion of patients with SIgED presented with duodenal ulcers (DU) (63.2% vs. 11.7%), who were positive for *Helicobacter pylori* (Hp) infection (47.4% vs. 11.7%) [98]. In our patients, the prevalence of DU was much lower (two patients, 3.8%, both positive for HP), which concurs with previous studies [23], including one from the group of Magen et al. [25], which did not find that SIgED can predispose to DU. The reasons that could explain the striking difference between their own two studies [25,98] are unclear and were not analyzed by the authors. 

The number of patients in whom serum levels of IgG subclasses had been assessed was small and prevented us from evaluating the possible clinical impact when both deficiencies concur.

Currently, very little is known about the mechanisms responsible for the deficiency in IgE, either in isolation or associated with deficiencies in other immunoglobulins. Similarly, the link between low IgE and the high risk of developing malignancy has yet to be elucidated. No abnormalities in the mechanisms involved in IgE synthesis have been reported so far in patients with SIgED. One study looked at the gene encoding activation-induced cytidine deaminase, which is an enzyme involved in immunoglobulin class switching, but the researchers could not find any mutation in patients with SIgED [99].

Our study has several limitations, such as the small number of patients recruited and its retrospective nature. We cannot exclude that some bias may have influenced our results. For example, the high prevalence of CSU and eczema found in our patients may be due to the fact that they are diseases treated by dermatologists and allergists who often include IgE measurement in their routine work-up. Moreover, we chose to use a stringent diagnostic criterion (IgE <2.0 kU/L) to increase the specificity in the diagnosis of SIgED. However, it remains unclear whether patients with IgE close to this level could also carry a similarly increased risk of developing diseases. There should be further studies gathering clinical data with different stratified IgE levels before a definitive serum IgE level can be established as a diagnostic threshold immunodeficient risk. Studies are also necessary to elucidate the clinical impact of complementary immunodeficient profiles such as associated IgG subtypes and the response to vaccines.

In summary, the data reported to date suggest that SIgED is characterized by a high prevalence of recurrent respiratory infections, asthma, autoimmune diseases, and malignancies [23,24,25,26]. Our study expands the spectrum of diseases associated with SIgED by adding bronchiectasis, enteropathy, CSU, eczematous dermatitis, LID, and liver diseases (PSC and hepatitis) to the known list. Although it is not clear why these diseases were not detected in previous studies, it is likely that the discrepancies are due to differences in the method used to recruit patients. In previous studies, patients were selected from allergy services [23,26] or with allergy-related symptoms [25], while in ours, patients were recruited without bias from the general hospital base, which could explain the higher prevalence of diseases that are not usually treated in the allergy units. Prospective studies based on broader populations are needed to further examine the role of SIgED in the development of different pathologies usually associated with immunodeficiencies. The possible genetic basis of SIgED is currently unknown and remains to be investigated. Hopefully, these studies will reveal whether SIgED can be added to the current list of antibody deficiencies.

## Figures and Tables

**Figure 1 jcm-10-04160-f001:**
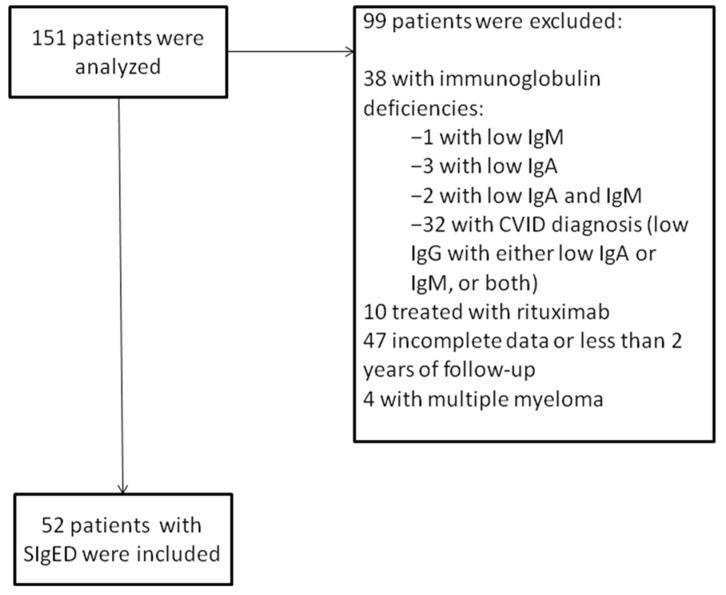
Flow chart of patients analyzed.

**Table 1 jcm-10-04160-t001:** Non-infectious complications.

	No. (%)
**Airway/Lung diseases**	
Bronchiectasis	16 (30.7)
Asthma	10 (19.2)
Rhinitis	8 (15.3)
Lymphocytic interstitial lung disease *	2 (3.8)
Interstitial lung fibrosis	1 (1.9)
**Autoimmune diseases**	
Hashimoto’s disease	8 (15.4)
Arthritis (1 RA, 4 undifferentiated)	5 (9.6)
Thrombopenia (2 associated with neutropenia)	4 (7.7)
Neutropenia	3 (5.8)
Aphthous stomatitis	3 (5.8)
Graves’ disease	2 (3.8)
Vitiligo	2 (3.8)
SLE	2 (3.8)
Alopecia	1 (1.9)
Acute hepatitis	1 (1.9)
Sjögren’s syndrome	1 (1.9)
**Gastrointestinal and liver diseases**	
Symptoms of enteropathy **	11 (21)
PSC	2 (3.8)
Ulcerative colitis (associated to PSC)	2 (3.8)
Cirrhosis (secondary to PSC)	2 (3.8)
**Other manifestations**	
Chronic spontaneous urticaria	9 (17.3)
Eczematous dermatitis	8 (15.3)
Polyarthralgia	9 (17.3)
Fatigue	7 (13.4)
Arterial hypertension	13 (33.3)

PSC, primary sclerosing cholangitis; RA, rheumatoid arthritis; SLE, systemic lupus erythematous. * Diagnosis based on CT scan images; Bronchoalveolar lavage lung fluid findings and biopsy of a lung nodule. ** Chronic or intermittent diarrhea, abdominal pain and bloating.

**Table 2 jcm-10-04160-t002:** Lymphomas and other tumors.

	No. (%)
**Lymphomas and Leukemias**	
Diffuse Large B cell lymphoma	3
Follicular cell lymphoma	1
Burkitt lymphoma	1
Lymphocytic Lymphoma/CLL	1
Lymphoma B cell, not otherwise specified	1
CLL	1
**Other**	
Melanoma	3
Breast	2
Skin cancer (basal cell carcinoma)	2
Cholangiocarcinoma	1
Hepatocarcinoma	1
Gynaecological (endometrial carcinoma)	1
Clear cell renal carcinoma	1
Meningioma	1
Neurinoma	1

CCL = Chronic Lymphocytic Leukemia.

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
