# Peer review of "Spectrum of Disease Manifestations in Patients with Selective Immunoglobulin E Deficiency"

_jcm, 2021, doi:10.3390/jcm10184160_

Round 1

Reviewer 1 Report

The authors summarize that the discrepancies between the current and previous epidemiological studies are due to differences in the method used to recruit patients. They wrote, "in previous studies [23–26], patients were mostly selected from allergy services, while in ours, patients were recruited without bias from the general hospital base, which could explain the higher prevalence of diseases that are not usually treated in the allergy units".  This statement is not correct, because some of the previous studies are based on the big database including the general population (Magen E, Schlesinger M, David M, Ben-Zion I, Vardy D. Selective IgE deficiency, immune dysregulation, and autoimmunity. Allergy Asthma Proc 2014; 35: e27-e33.)

Moreover, the population with selective igE deficiency in this study was retrospectively recruited from the hospital-based data, Therefore this is the major limitation of this study, which should be mentioned among the study limitations. The population selected from the hospital database might provide a serious bias, because it may not fully represent the all ambulatory + asymptomatic population with selective IgE deficiency. 

Reviewer 2 Report

This is an interesting study on selective IgE deficiency (SIgED). However, the manuscript as currently written has some inaccuracies and is too long.

Introduction

The manuscript states that "SIgED has been previously evaluated in selected patients from allergy units…". This is correct for the first 2 studies on this matter (refs 23 and 24 in the manuscript), yet this is incorrect for the last study quoted (reference 25). The first 2 studies were indeed performed utilizing patients from allergy referrals and had a small number of participants, The study by Magen et al (ref 25) was performed by screening electronic health records of a large health service, selecting those who had selective IgE deficiency and searching the diagnosis data bases for SIgED patients compared to non-SIgE patients. Their results are similar to yours. In addition, the same group also published 2 additional studies on SIgED patients using the same methods. They showed that Arterial hypertension/coronary artery disease/cerebrovascular disease and Helicobacter associated gastritis are also more common in SIgED patients (Magen E, Mishal J, Vardy D. Selective IgE deficiency and cardiovascular diseases. Allergy Asthma Proc. 2015;36(3):225-229. doi:10.2500/aap.2015.36.3825, (Magen E, Mishal J, Vardy D. Selective IgE deficiency and cardiovascular diseases. Allergy Asthma Proc. 2015 May-Jun;36(3):225-9. doi: 10.2500/aap.2015.36.3825. PMID: 25976439; PMCID: PMC4405604.) I would add these references.

I suggest shortening the introduction to an explanation that IgE deficiency is very common in immunodeficiency diseases that affect more than one class of immunoglobulin and then move on to a short review of previous studies on SIgED. I would delete the sentence in page 2 lines 91-93. SIgED has not been "overlooked until now" as stated in your manuscript but I do agree that it has not been emphasized enough as a marker for immunodeficiency/autoimmunity.

In the last paragraph of the introduction I would state that this study is the first to research the effects of SIgED in the entire population in a hospital setting with 2 years of follow up and this study sought to delineate in detail the clinical aspects of SIgED.

Methods

I would first state the inclusion criteria for the study. Something like "Any patient who was found to have an IgE concentration below 2 kU/L with normal IgG, IgM and IgA concentrations for whom at least 2 years follow up at our institution was available". Figure 1 should be deleted. I would also delete table 1 and state the normal ranges in the text (adding the results of the patients in table 2 is part of the results and not methods).

Results

I would add a table for infectious complications so that there will be 3 tables (Infectious complications, Autoimmune/non Infectious complications ( Bronchiectasis is currently listed as non-infectious, consider listing this in the infectious complications and adding "after recurrent pneumonia") and malignancy.

I would also add a table for how many patients had combinations of the presentations (i.e- How many patients had recurrent pneumonia, bronchiectasis, autoimmunity and lymphomas or a combination of these).

Discussion

The discussion is too long and should be shortened. As presented it is very difficult to read (4 pages). I suggest writing your main findings first and then showing how your results expand what is known about SIgED and what you have added stressing that your methods (hospital setting-entire hospital population) and 2 years follow up are unique. A 2 page discussion will be easier to read and understand.

I agree with your assumption that genetic studies may reveal new disease or diseases associated with SIgED that have clinical relevance. Are you performing a study in this direction? It is possible that your patients may reveal such a disease.

Reviewer 3 Report

The authors collected 52 patients lacking IgE but not whole IgG, IgA, or IgM retrospectively and analyzed their clinical manifestations. However, the IgG sublcasses are measured only in 14 patients, of which three showed IgG subclass deficiency. In addition, no IgA subclasses (i.e., IgA1 and IgA2) were analyzed. Although the data shown are valuable in some clinical settings, these points limit the etiological or diagnostic meanings of the report, since the clinical manifestations are similar to CVID.

Major points:
1. Since the data of the subclasses of IgG and IgA are largely missing, the title "Selective Immunoglobulin E deficiency" is not accurate. Because IgA2 is a neighbor of IgE locus, the diagnosis may contain some IgA2 deficient patients. This may explain the manifestations of bronchiectasis and enteropathy, where the secreted type of IgAs (mainly IgA2) play a role. In addition to correcting the title, this limitation should be made explicit in the abstract.
2. Since IgG subclass deficiency is known to affect the clinical manifestations, the clinical manifestations of 11 patients without IgG subclass deficiency should be described.
3. Why IgE amount was measured in the patients included? That should be a selection bias.

Round 2

Reviewer 2 Report

The manuscript has improved, although I still believe the manuscript is too long. As presented, the manuscript is very detailed which makes it a good reference for future studies on SIgE deficiency; however, it is very difficult to read. If the authors wanted to create a detailed reference to every aspect of selective IgE deficiency, they have indeed succeeded, yet, it is tedious. If the authors want to write a manuscript that is "reader friendly" they did not succeed in that.

One comment was still not addressed in the revised manuscript: Did your SIgE deficiency patients have Helicobacter associated gastritis? Currently your study does not have even one patient with this diagnosis, yet in another manuscript by Magen et al, Helicobacter associated gastritis was very common and significantly more prevalent than patients without SIgE deficiency (Magen E, Schlesinger M, Ben-Zion I, Vardy D. Helicobacter pylori infection in patients with selective immunoglobulin E deficiency. World J Gastroenterol. 2015 Jan 7;21(1):240-5. doi: 10.3748/wjg.v21.i1.240. PMID: 25574097; PMCID: PMC4284341). If you have data regarding Helicobacter associated gastritis, please add these data, if not, I would still comment on this.

Reviewer 3 Report

Although the authors rebutted faithfully to the raised concerns, my concerns were not resolved in the two points below.

  1. Reviewer. Since IgG subclass deficiency is known to affect the clinical manifestations, the clinical manifestations of 11 patients without IgG subclass deficiency should be described.

Authors. We honestly believe that a scientific or clinically relevant contribution can not be made with the mere description of a small number of patients without comparing them with a group of patients with subclasses deficiencies.

Reviewer: I believe the authors misunderstood my comment. Because most of the comorbidities they found in IgE deficiency patients were more than 10% in frequency, the 11 patients confirmed to be “without IgG subclass deficiency” patients (out of 52 included) should recapitulate the clinical picture of entire population of the apparent specific IgE deficiency. On the contrary, if the 11 did not show any comorbidities, that suggest the clinicians should test IgG subclass when they encounter extremely low IgE. Even if the latter is the case, it does not damage the value of this study. The information should be described, even if not necessarily in detail.

  1. Reviewer. Why IgE amount was measured in the patients included? That should be a selection bias

Authors. IgE was studied in most cases due to suspected allergic origin of respiratory diseases (rhinitis, asthma), bronchiectasis (suspected bronchopulmonary aspergillosis), cutaneous (chronic spontaneous urticaria, eczema), digestive (food allergy),and adverse reaction to drugs. In some cases the analysis was routine work up without clear motivation. As expected allergy tests (skin prick test and specific IgE against the suspected culprit agents) were negative in all patients with suspected allergy.

The possibility of bias selection in the determination of IgE is recognized in the list of the limitations of the study described in the manuscript. In any case, the possible existence of a selection bias can hardly explain the high frequency of autoimmune, malignant tumours, liver, and rheumatic diseases found in the patients.

Reviewer: The authors described that they obtained the data “without selection bias” in the abstract. This should be rephrased to “consulting any departments” or equivalents.
